# Make Sharpness-Aware Minimization Stronger: A Sparsified Perturbation Approach

**Peng Mi**[1][*]   **Li Shen**[2][†]   **Tianhe Ren**[1]   **Yiyi Zhou**[1]
**Xiaoshuai Sun**[1]   **Rongrong Ji**[1]   **Dacheng Tao**[2,3]

[1]Media Analytics and Computing Laboratory, Department of Artificial Intelligence,
School of Informatics, Xiamen University, China
[2]JD Explore Academy, Beijing, China   [3]The University of Sydney, Australia
mipeng@stu.xmu.edu.cn, mathshenli@gmail.com, rentianhe@stu.xmu.edu.cn
zhouyiyi@xmu.edu.cn, xssun@xmu.edu.cn
rrji@xmu.edu.cn, dacheng.tao@gmail.com

## Abstract

Deep neural networks often suffer from poor generalization caused by complex and non-convex loss landscapes. One of the popular solutions is Sharpness-Aware Minimization (SAM), which smooths the loss landscape via minimizing the maximized change of training loss when adding a perturbation to the weight. However, we find the indiscriminate perturbation of SAM on all parameters is suboptimal, which also results in excessive computation, *i.e.*, double the overhead of common optimizers like Stochastic Gradient Descent (SGD). In this paper, we propose an efficient and effective training scheme coined as Sparse SAM (SSAM), which achieves sparse perturbation by a binary mask. To obtain the sparse mask, we provide two solutions which are based on Fisher information and dynamic sparse training, respectively. In addition, we theoretically prove that SSAM can converge at the same rate as SAM, *i.e.*, $O(\log T/\sqrt{T})$. Sparse SAM not only has the potential for training acceleration but also smooths the loss landscape effectively. Extensive experimental results on CIFAR10, CIFAR100, and ImageNet-1K confirm the superior efficiency of our method to SAM, and the performance is preserved or even better with a perturbation of merely 50% sparsity. Code is available at https://github.com/Mi-Peng/Sparse-Sharpness-Aware-Minimization.

## 1   Introduction

Over the past decade or so, the great success of deep learning has been due in great part to ever-larger model parameter sizes [13, 56, 52, 10, 40, 5]. However, the excessive parameters also make the model inclined to poor generalization. To overcome this problem, numerous efforts have been devoted to training algorithm [24, 42, 50], data augmentation [11, 57, 55], and network design [26, 28].

One important finding in recent research is the connection between the geometry of loss landscape and model generalization [31, 17, 27, 44, 53]. In general, the loss landscape of the model is complex and non-convex, which makes model tend to converge to sharp minima. Recent endeavors [31, 27, 44] show that the flatter the minima of convergence, the better the model generalization. This discovery reveals the nature of previous approaches [24, 26, 11, 55, 57, 28] to improve generalization, *i.e.*, smoothing the loss landscape.

---

[*]This work was done during an internship at JD Explore Academy.
[†]Li Shen is the corresponding author.

36th Conference on Neural Information Processing Systems (NeurIPS 2022).

Based on this finding, Foret *et al.* [17] propose a novel approach to improve model generalization called *sharpness-aware minimization* (SAM), which simultaneously minimizes loss value and loss sharpness. SAM quantifies the landscape sharpness as the maximized difference of loss when a perturbation is added to the weight. When the model reaches a sharp area, the perturbed gradients in SAM help the model jump out of the sharp minima. In practice, SAM requires two forward-backward computations for each optimization step, where the first computation is to obtain the perturbation and the second one is for parameter update. Despite the remarkable performance [17, 34, 14, 7], This property makes SAM double the computational cost of the conventional optimizer, *e.g.*, SGD [3].

Since SAM calculates perturbations indiscriminately for all parameters, a question is arisen:

*Do we need to calculate perturbations for all parameters?*

Above all, we notice that in most deep neural networks, only about 5% of parameters are sharp and rise steeply during optimization [31]. Then we explore the effect of SAM in different dimensions to answer the above question and find out *(i) little difference between SGD and SAM gradients in most dimensions (see Fig. 1); (ii) more flatter without SAM in some dimensions (see Fig. 4 and Fig. 5).*

Inspired by the above discoveries, we propose a novel scheme to improve the efficiency of SAM via sparse perturbation, termed Sparse SAM (SSAM). SSAM, which plays the role of regularization, has better generalization, and its sparse operation also ensures the efficiency of optimization. Specifically, the perturbation in SSAM is multiplied by a binary sparse mask to determine which parameters should be perturbed. To obtain the sparse mask, we provide two implementations. The first solution is to use Fisher information [16] of the parameters to formulate the binary mask, dubbed SSAM-F. The other one is to employ dynamic sparse training to jointly optimize model parameters and the sparse mask, dubbed SSAM-D. The first solution is relatively more stable but a bit time-consuming, while the latter is more efficient.

In addition to these solutions, we provide the theoretical convergence analysis of SAM and SSAM in non-convex stochastic setting, proving that our SSAM can converge at the same rate as SAM, *i.e.*, $O(\log T/\sqrt{T})$. At last, we evaluate the performance and effectiveness of SSAM on CIFAR10 [33], CIFAR100 [33] and ImageNet [8] with various models. The experiments confirm that SSAM contributes to a flatter landscape than SAM, and its performance is on par with or even better than SAM with only about 50% perturbation. These results coincide with our motivations and findings.

To sum up, the contribution of this paper is three-fold:

- We rethink the role of perturbation in SAM and find that the indiscriminate perturbations are suboptimal and computationally inefficient.
- We propose a sparsified perturbation approach called Sparse SAM (SSAM) with two variants, *i.e.*, Fisher SSAM (SSAM-F) and Dynamic SSAM (SSAM-D), both of which enjoy better efficiency and effectiveness than SAM. We also theoretically prove that SSAM can converge at the same rate as SAM, *i.e.*, $O(\log T/\sqrt{T})$.
- We evaluate SSAM with various models on CIFAR and ImageNet, showing WideResNet with SSAM of a high sparsity outperforms SAM on CIFAR; SSAM can achieve competitive performance with a high sparsity; SSAM has a comparable convergence rate to SAM.

## 2 Related Work

In this section, we briefly review the studies on sharpness-aware minimum optimization (SAM), Fisher information in deep learning, and dynamic sparse training.

**SAM and flat minima.** Hochreiter *et al.* [27] first reveal that there is a strong correlation between the generalization of a model and the flat minima. After that, there is a growing amount of research based on this finding. Keskar *et al.* [31] conduct experiments with a larger batch size, and in consequence observe the degradation of model generalization capability. They [31] also confirm the essence of this phenomenon, which is that the model tends to converge to the sharp minima. Keskar *et al.* [31] and Dinh *et al.* [12] state that the sharpness can be evaluated by the eigenvalues of the Hessian. However, they fail to find the flat minima due to the notorious computational cost of Hessian.

Inspired by this, Foret *et al.* [17] introduce a sharpness-aware optimization (SAM) to find a flat minimum for improving generalization capability, which is achieved by solving a mini-max problem.

Zhang *et al.* [58] make a point that SAM [17] is equivalent to adding the regularization of the gradient norm by approximating Hessian matrix. Kwon *et al.* [34] propose a scale-invariant SAM scheme with adaptive radius to improve training stability. Zhang *et al.* [59] redefine the landscape sharpness from an intuitive and theoretical perspective based on SAM. To reduce the computational cost in SAM, Du *et al.* [14] proposed Efficient SAM (ESAM) to randomly calculate perturbation. However, ESAM randomly select the samples every steps, which may lead to optimization bias. Instead of the perturbations for all parameters, *i.e.*, SAM, we compute a sparse perturbation, *i.e.*, SSAM, which learns important but sparse dimensions for perturbation.

**Fisher information (FI).** Fisher information [16] was proposed to measure the information that an observable random variable carries about an unknown parameter of a distribution. In machine learning, Fisher information is widely used to measure the importance of the model parameters [32] and decide which parameter to be pruned [49, 51]. For proposed SSAM-F, Fisher information is used to determine whether a weight should be perturbed for flat minima.

**Dynamic sparse training.** Finding the sparse network via pruning unimportant weights is a popular solution in network compression, which can be traced back to decades [35]. The widely used training scheme, *i.e.*, pretraining-pruning-fine-tuning, is presented by Han *et.al.* [23]. Limited by the requirement for the pre-trained model, some recent research [15, 2, 9, 30, 43, 37, 38] attempts to discover a sparse network directly from the training process. Dynamic Sparse Training (DST) finds the sparse structure by dynamic parameter reallocation. The criterion of pruning could be weight magnitude [18], gradient [15] and Hessian [35, 48], *etc*. We claim that different from the existing DST methods that prune neurons, our target is to obtain a binary mask for sparse perturbation.

## 3 Rethinking the Perturbation in SAM

In this section, we first review how SAM converges at the flat minimum of a loss landscape. Then, we rethink the role of perturbation in SAM.

### 3.1 Preliminary

In this paper, we consider the weights of a deep neural network as $\boldsymbol{w} = (w_1, w_2, ..., w_d) \subseteq \mathcal{W} \in \mathbb{R}^d$ and denote a binary mask as $\boldsymbol{m} \in \{0, 1\}^d$, which satisfies $\mathbf{1}^T \boldsymbol{m} = (1-s) \cdot d$ to restrict the computational cost. Given a training dataset as $\mathcal{S} \triangleq \{(\boldsymbol{x}_i, \boldsymbol{y}_i)\}_{i=1}^n$ *i.i.d.* drawn from the distribution $\mathcal{D}$, the per-data-point loss function is defined by $f(\boldsymbol{w}, \boldsymbol{x}_i, \boldsymbol{y}_i)$. For the classification task in this paper, we use cross-entropy as loss function. The population loss is defined by $f_{\mathcal{D}} = \mathbb{E}_{(\boldsymbol{x}_i, \boldsymbol{y}_i) \sim \mathcal{D}} f(\boldsymbol{w}, \boldsymbol{x}_i, \boldsymbol{y}_i)$, while the empirical training loss function is $f_{\mathcal{S}} \triangleq \frac{1}{n} \sum_{i=1}^n f(\boldsymbol{w}, \boldsymbol{x}_i, \boldsymbol{y}_i)$.

Sharpness-aware minimization (SAM) [17] aims to simultaneously minimize the loss value and smooth the loss landscape, which is achieved by solving the min-max problem:

$$\min_{\boldsymbol{w}} \max_{||\boldsymbol{\epsilon}||_2 \leq \rho} f_{\mathcal{S}}(\boldsymbol{w} + \boldsymbol{\epsilon}). \tag{1}$$

SAM first obtains the perturbation $\boldsymbol{\epsilon}$ in a neighborhood ball area with a radius denoted as $\rho$. The optimization tries to minimize the loss of the perturbed weight $\boldsymbol{w} + \boldsymbol{\epsilon}$. Intuitively, the goal of SAM is that small perturbations to the weight will not significantly rise the empirical loss, which indicates that SAM tends to converge to a flat minimum. To solve the mini-max optimization, SAM approximately calculates the perturbations $\boldsymbol{\epsilon}$ using Taylor expansion around $\boldsymbol{w}$:

$$\boldsymbol{\epsilon} = \arg\max_{||\boldsymbol{\epsilon}||_2 \leq \rho} f_{\mathcal{S}}(\boldsymbol{w} + \boldsymbol{\epsilon}) \approx \arg\max_{||\boldsymbol{\epsilon}||_2 \leq \rho} f_{\mathcal{S}}(\boldsymbol{w}) + \boldsymbol{\epsilon} \cdot \nabla_{\boldsymbol{w}} f(\boldsymbol{w}) = \rho \cdot \nabla_{\boldsymbol{w}} f(\boldsymbol{w}) / ||\nabla_{\boldsymbol{w}} f(\boldsymbol{w})||_2. \tag{2}$$

In this way, the objective function can be rewritten as $\min_{\boldsymbol{w}} f_{\mathcal{S}}(\boldsymbol{w} + \rho \nabla_{\boldsymbol{w}} f(\boldsymbol{w}) / ||\nabla_{\boldsymbol{w}} f(\boldsymbol{w})||_2)$, which could be implemented by a two-step gradient descent framework in Pytorch or TensorFlow:

- In the first step, the gradient at $\boldsymbol{w}$ is used to calculate the perturbation $\boldsymbol{\epsilon}$ by Eq. 2. Then the weight of model will be added to $\boldsymbol{w} + \boldsymbol{\epsilon}$.

- In the second step, the gradient at $\boldsymbol{w} + \boldsymbol{\epsilon}$ is used to solve $\min_{\boldsymbol{w}} f_{\mathcal{S}}(\boldsymbol{w} + \boldsymbol{\epsilon})$, *i.e.*, update the weight $\boldsymbol{w}$ by this gradient.

## 3.2  Rethinking the Perturbation Step of SAM

**How does SAM work in flat subspace?**  SAM perturbs all parameters indiscriminately, but the fact is that merely about 5% parameter space is sharp while the rest is flat [31]. We are curious whether perturbing the parameters in those already flat dimensions would lead to the instability of the optimization and impair the improvement of generalization. To answer this question, we quantitatively and qualitatively analyze the loss landscapes with different training schemes in Section 5, as shown in Fig. 4 and Fig. 5. The results confirm our conjecture that optimizing some dimensions without perturbation can help the model generalize better.

**What is the difference between the gradients of SGD and SAM?**  We investigate various neural networks optimized with SAM and SGD on CIFAR10/100 and ImageNet, whose statistics are given in Fig. 1. We use the relative difference ratio $r$, defined as $r = \log |(g_{SAM} - g_{SGD})/g_{SGD}|$, to measure the difference between the gradients of SAM and SGD. As showin in Fig. 1, the parameters with $r$ less than 0 account for the vast majority of all parameters, indicating that most SAM gradients are not significantly different from SGD gradients. These results show that most parameters of the model require no perturbations for achieving the flat minima, which well confirms our the motivation.

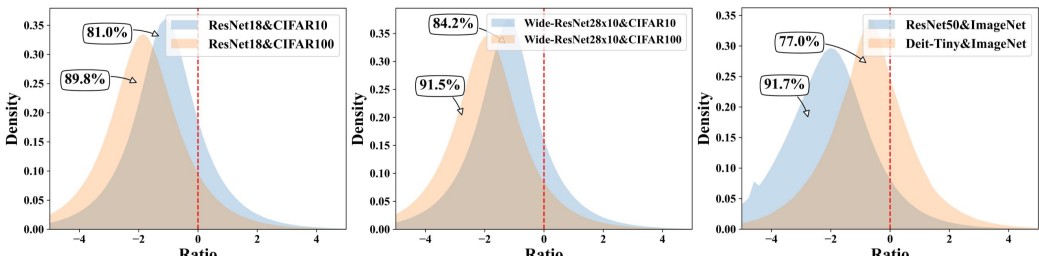

Figure 1: The distribution of relative difference ratio $r$ among various models and datasets. There is little difference between SAM and SGD gradients for most parameters, *i.e.*, the ratio $r$ is less than 0.

Inspired by the above observation and the promising hardware acceleration for sparse operation modern GPUs, we further propose Sparse SAM, a novel sparse perturbation approach, as an implicit regularization to improve the efficiency and effectiveness of SAM.

## 4  Methodology

In this section, we first define the proposed Sparse SAM (SSAM), which strengths SAM via sparse perturbation. Afterwards, we introduce the instantiations of the sparse mask used in SSAM via Fisher information and dynamic sparse training, dubbed SSAM-F and SSAM-D, respectively.

### 4.1  Sparse SAM

Motivated by the finding discussed in the introduction, Sparse SAM (SSAM) employs a sparse binary mask to decide which parameters should be perturbed, thereby improving the efficiency of sharpness-aware minimization. Specifically, the perturbation $\epsilon$ will be multiplied by a sparse binary mask $\boldsymbol{m}$, and the objective function is then rewritten as $\min_{\boldsymbol{w}} f_{\mathcal{S}} \left( \boldsymbol{w} + \rho \cdot \frac{\nabla_{\boldsymbol{w}} f(\boldsymbol{w})}{||\nabla_{\boldsymbol{w}} f(\boldsymbol{w})||_2} \odot \boldsymbol{m} \right)$. To stable the optimization, the sparse binary mask $\boldsymbol{m}$ is updated at regular intervals during training. We provide two solutions to obtain the sparse mask $\boldsymbol{m}$, namely Fisher information based Sparse SAM (SSAM-F) and dynamic sparse training based Sparse SAM (SSAM-D). The overall algorithms of SSAM and sparse mask generations are shown in Algorithm 1 and Algorithm 2, respectively.

According to the previous work [54] and Ampere architecture equipped with sparse tensor cores [47, 46, 45], currently there exists technical support for matrix multiplication with 50% fine-grained sparsity [54][3]. Therefore, SSAM of 50% sparse perturbation has great potential to achieve true training acceleration via sparse back-propagation.

---

[3]For instance, 2:4 sparsity for A100 GPU.

| **Algorithm 1** Sparse SAM (SSAM) | **Algorithm 2** Sparse Mask Generation |
|---|---|

**Algorithm 1** Sparse SAM (SSAM)

**Input:** sparse ratio $s$, dense model $\boldsymbol{w}$, binary mask $\boldsymbol{m}$, update interval $T_m$, number of samples $N_F$, learning rate $\eta$, training set $\mathcal{S}$.
1: Initialize $\boldsymbol{w}$ and $\boldsymbol{m}$ randomly.
2: **for** epoch $t = 1, 2 \ldots T$ **do**
3:    **for** each training iteration **do**
4:       Sample a batch from $\mathcal{S}$: $\mathcal{B}$
5:       Compute perturbation $\boldsymbol{\epsilon}$ by Eq. (2)
6:       **if** $t \mod T_m = 0$ **then**
7:          Generate mask $m$ via Option I or II
8:       **end if**
9:       $\boldsymbol{\epsilon} \leftarrow \boldsymbol{\epsilon} \odot \boldsymbol{m}$
10:    **end for**
11:    $\boldsymbol{w} \leftarrow \boldsymbol{w} - \eta \nabla f(\boldsymbol{w} + \boldsymbol{\epsilon})$
12: **end for**
13: **return** Final weight of model $\boldsymbol{w}$

**Algorithm 2** Sparse Mask Generation

1: Option I:(Fisher Information Mask)
2: Sample $N_F$ data from $\mathcal{S}$: $\mathcal{B}_F$
3: Compute Fisher $\hat{F}_{\boldsymbol{w}}$ by Eq. (5)
4: $\boldsymbol{m_1} = \{m_i = 1 | m_i \in \boldsymbol{m}\} \leftarrow \mathrm{ArgTopK}(\hat{F}_{\boldsymbol{w}}, s \cdot |\boldsymbol{w}|)$
5: $\boldsymbol{m_0} = \{m_i = 0 | m_i \in \boldsymbol{m}\} \leftarrow \{m_i | m_i \notin \boldsymbol{m_1}\}$
6: $\boldsymbol{m} \leftarrow \boldsymbol{m_0} \cup \boldsymbol{m_1}$
7: Option II:(Dynamic Sparse Mask)
8: $N_{drop} = f_{decay}(t; \alpha) \cdot (1 - s) \cdot |\boldsymbol{w}|$
9: $N_{growth} = N_{drop}$
10: $\boldsymbol{m_1} = \{m_i = 1 | m_i \in \boldsymbol{m}\} \leftarrow \{m_i = 1 | m_i \in \boldsymbol{m}\} - \mathrm{ArgTopK}_{m_i \in \boldsymbol{m_1}}(-|\nabla f(\boldsymbol{w})|, N_{drop})$
11: $\boldsymbol{m_1} \leftarrow \{m_i = 1 | m_i \in \boldsymbol{m}\} + \mathrm{Random}_{m_i \notin \boldsymbol{m_1}}(N_{growth})$
12: $\boldsymbol{m_0} = \{m_i = 0 | m_i \in \boldsymbol{m}\} \leftarrow \{m_i | m_i \notin \boldsymbol{m_1}\}$
13: $\boldsymbol{m} \leftarrow \boldsymbol{m_0} \cup \boldsymbol{m_1}$
14: **return** Sparse mask $\boldsymbol{m}$

## 4.2 SSAM-F: Fisher information based Sparse SAM

Inspired by the connection between Fisher information and Hessian [16], which can directly measure the flatness of the loss landscape, we apply Fisher information to achieve sparse perturbation, denoted as SSAM-F. The Fisher information is defined by

$$F_{\boldsymbol{w}} = \mathbb{E}_{x \sim p(x)} \left[ \mathbb{E}_{y \sim p_{\boldsymbol{w}}(y|x)} \nabla_{\boldsymbol{w}} \log p_{\boldsymbol{w}}(y|x) \nabla_{\boldsymbol{w}} \log p_{\boldsymbol{w}}(y|x)^T \right], \tag{3}$$

where $p_{\boldsymbol{w}}(y|x)$ is the output distribution predicted by the model. In over-parameterized networks, the computation of Fisher information matrix is also intractable, *i.e.*, $|\boldsymbol{w}| \times |\boldsymbol{w}|$. Following [49, 19, 41, 22], we approximate $F_{\boldsymbol{w}}$ as a diagonal matrix, which is included in a vector in $\mathbb{R}^{|\boldsymbol{w}|}$. Note that there are the two expectation in Eq. (3). The first one is that the original data distribution $p(x)$ is often not available. Therefore, we approximate it by sampling $N_F$ data $\boldsymbol{x}_1, \boldsymbol{x}_2, \ldots, \boldsymbol{x}_{N_F}$ from $x \sim p(x)$:

$$F_{\boldsymbol{w}} = \frac{1}{N_F} \mathbb{E}_{y \sim p_{\boldsymbol{w}}(y|\boldsymbol{x}_i)} (\nabla_{\boldsymbol{w}} \log p_{\boldsymbol{w}}(y|\boldsymbol{x}_i))^2. \tag{4}$$

For the second expectation over $p_{\boldsymbol{w}}(y|\boldsymbol{x})$, it is not necessary to compute its explicit expectation, since the ground-truth $y_i$ for each training sample $x_i$ is available in supervised learning. Therefore we rewrite the Eq. (4) as "empirical Fisher":

$$\hat{F}_{\boldsymbol{w}} = \frac{1}{N_F} (\nabla_{\boldsymbol{w}} \log p_{\boldsymbol{w}}(y_i|\boldsymbol{x}_i))^2. \tag{5}$$

We emphasize that the empirical Fisher is a $|\boldsymbol{w}|$-dimension vector, which is the same as the mask $\boldsymbol{m}$. To obtain the mask $\boldsymbol{m}$, we calculate the empirical Fisher by Eq. (5) over $N_F$ training data randomly sampled from training set $\mathcal{S}$. Then we sort the elements of empirical Fisher in descending, and the parameters corresponding to the the top $k$ Fisher values will be perturbed:

$$\boldsymbol{m_1} = \{m_i = 1 | m_i \in \boldsymbol{m}\} \leftarrow \mathrm{ArgTopK}(\hat{F}_{\boldsymbol{w}}, k), \tag{6}$$

where $\mathrm{ArgTopK}(\boldsymbol{v}, N)$ returns the index of the top $N$ largest values among $\boldsymbol{v}$, $\boldsymbol{m_1}$ is the set of values that are 1 in $\boldsymbol{m}$. $k$ is the number of perturbed parameters, which is equal to $(1 - s) \cdot |\boldsymbol{w}|$ for sparsity $s$. After setting the rest values of the mask to 0, *i.e.*, $\boldsymbol{m_0} = \{m_i = 0 | m_i \notin \boldsymbol{m_1}\}$, we get the final mask $\boldsymbol{m} = \boldsymbol{m_0} \cup \boldsymbol{m_1}$. The algorithm of SSAM-F is shown in Algorithm 2.

## 4.3 SSAM-D: Dynamic sparse training mask based sparse SAM

Considering the computation of empirical Fisher is still relatively high, we also resort to dynamic sparse training for efficient binary mask generation. The mask generation includes the perturbation dropping and the perturbation growth steps. At the perturbation dropping phase, the flattest dimensions of the perturbed parameters will be dropped, *i.e.*, the gradients of lower absolute values, which means that they require no perturbations. The update of the sparse mask follows

$$\boldsymbol{m_1} = \{m_i = 1 | m_i \in \boldsymbol{m}\} \leftarrow \boldsymbol{m_1} - \mathrm{ArgTopK}_{\boldsymbol{w} \in \boldsymbol{m_1}}(-|\nabla f(\boldsymbol{w})|, N_{drop}), \tag{7}$$

where $N_{drop}$ is the number of perturbations to be dropped. At the perturbation growth phase, for the purpose of exploring the perturbation combinations as many as possible, several unperturbed dimensions grow, which means these dimensions need to compute perturbations. The update of the sparse mask for perturbation growth follows

$$\boldsymbol{m_1} = \{m_i = 1 | m_i \in \boldsymbol{m}\} \leftarrow \boldsymbol{m_1} + \text{Random}_{\boldsymbol{w} \notin \boldsymbol{m_1}}(N_{growth}), \tag{8}$$

where $\text{Random}_{\mathcal{S}}(N)$ randomly returns $N$ indexes in $\mathcal{S}$, and $N_{growth}$ is the number of perturbation growths. To keep the sparsity constant during training, the number of growths is equal to the number of dropping, i.e., $N_{growth} = N_{drop}$. Afterwards, we set the rest values of the mask to 0, i.e., $\boldsymbol{m_0} = \{m_i = 0 | m_i \notin \boldsymbol{m_1}\}$, and get the final mask $\boldsymbol{m} = \boldsymbol{m_0} \cup \boldsymbol{m_1}$. The drop ratio $\alpha$ represents the proportion of dropped perturbations in the total perturbations $s \cdot |\boldsymbol{w}|$, i.e., $\alpha = N_{drop}/(s \cdot |\boldsymbol{w}|)$. In particular, a larger drop rate means that more combinations of binary mask can be explored during optimization, which, however, may slightly interfere the optimization process. Following [15, 9], we apply a cosine decay scheduler to alleviate this problem:

$$f_{decay}(t; \alpha) = \frac{\alpha}{2} \left(1 + \cos\left(t\pi/T\right)\right), \tag{9}$$

where $T$ denotes number of training epochs. The algorithm of SSAM-D is depicted in Algorithm 2.

### 4.4 Theoretical analysis of Sparse SAM

In the following, we analyze the convergence of SAM and SSAM in non-convex stochastic setting. Before introducing the main theorem, we first describe the following assumptions that are commonly used for characterizing the convergence of nonconvex stochastic optimization [39, 59, 1, 6, 4, 20].

**Assumption 1.** *(Bounded Gradient.) It exists $G \geq 0$ s.t. $||\nabla f(\boldsymbol{w})|| \leq G$.*

**Assumption 2.** *(Bounded Variance.) It exists $\sigma \geq 0$ s.t. $\mathbb{E}[||g(\boldsymbol{w}) - \nabla f(\boldsymbol{w})||^2] \leq \sigma^2$.*

**Assumption 3.** *(L-smoothness.) It exists $L > 0$ s.t. $||\nabla f(\boldsymbol{w}) - \nabla f(\boldsymbol{v})|| \leq L||\boldsymbol{w} - \boldsymbol{v}||, \forall \boldsymbol{w}, \boldsymbol{v} \in \mathbb{R}^d$.*

**Theorem 1.** *Consider function $f(\boldsymbol{w})$ satisfying the Assumptions 1-3 optimized by SAM. Let $\eta_t = \frac{\eta_0}{\sqrt{t}}$ and perturbation amplitude $\rho$ decay with square root of $t$, e.g., $\rho_t = \frac{\rho_0}{\sqrt{t}}$. With $\rho_0 \leq G\eta_0$, we have*

$$\frac{1}{T} \sum_{t=0}^{T} \mathbb{E}||\nabla f(\boldsymbol{w}_t)||^2 \leq C_1 \frac{1}{\sqrt{T}} + C_2 \frac{\log T}{\sqrt{T}}, \tag{10}$$

*where $C_1 = \frac{2}{\eta_0}(f(\boldsymbol{w}_0) - \mathbb{E}f(\boldsymbol{w}_T))$ and $C_2 = 2(L\sigma^2\eta_0 + LG\rho_0)$.*

**Theorem 2.** *Consider function $f(\boldsymbol{w})$ satisfying the Assumptions 1-3 optimized by SSAM. Let $\eta_t = \frac{\eta_0}{\sqrt{t}}$ and perturbation amplitude $\rho$ decay with square root of $t$, e.g., $\rho_t = \frac{\rho_0}{\sqrt{t}}$. With $\rho_0 \leq G\eta_0/2$, we have:*

$$\frac{1}{T} \sum_{t=0}^{T} \mathbb{E}||\nabla f(\boldsymbol{w}_t)||^2 \leq C_3 \frac{1}{\sqrt{T}} + C_4 \frac{\log T}{\sqrt{T}}, \tag{11}$$

*where $C_3 = \frac{2}{\eta_0}(f(\boldsymbol{w}_0) - \mathbb{E}f(\boldsymbol{w}_T) + \eta_0 L^2 \rho^2 (1 + \eta_0 L)\frac{\pi^2}{6})$ and $C_4 = 2(L\sigma^2\eta_0 + LG\rho_0)$.*

For non-convex stochastic optimization, Theorems 1&2 imply that our SSAM could converge the same rate as SAM, i.e., $O(\log T/\sqrt{T})$. Detailed proofs of the two theorems are given in **Appendix**.

## 5 Experiments

In this section, we evaluate the effectiveness of SSAM through extensive experiments on CIFAR10, CIFAR100 [33] and ImageNet-1K [8]. The base models include ResNet [25] and WideResNet [56]. We report the main results on CIFAR datasets in Tables 1&2 and ImageNet-1K in Table 3. Then, we visualize the landscapes and Hessian spectra to verify that the proposed SSAM can help the model generalize better. More experimental results are placed in **Appendix** due to page limit.

## 5.1 Implementation details

**Datasets.** We use CIFAR10/CIFAR100 [33] and ImageNet-1K [8] as the benchmarks of our method. Specifically, CIFAR10 and CIFAR100 have 50,000 images of 32×32 resolution for training, while 10,000 images for test. ImageNet-1K [8] is the most widely used benchmark for image classification, which has 1,281,167 images of 1000 classes and 50,000 images for validation.

**Hyper-parameter setting.** For small resolution datasets, *i.e.*, CIFAR10 and CIFAR100, we replace the first convolution layer in ResNet and WideResNet with the one of 3×3 kernel size, 1 stride and 1 padding. The models on CIFAR10/CIFAR100 are trained with 128 batch size for 200 epochs. We apply the random crop, random horizontal flip, normalization and cutout [11] for data augmentation, and the initial learning rate is 0.05 with a cosine learning rate schedule. The momentum and weight decay of SGD are set to 0.9 and 5$e$-4, respectively. SAM and SSAM apply the same settings, except that weight decay is set to 0.001 [14]. We determine the perturbation magnitude $\rho$ from $\{0.01, 0.02, 0.05, 0.1, 0.2, 0.5\}$ via grid search. In CIFAR10 and CIFAR100, we set $\rho$ as 0.1 and 0.2, respectively. For ImageNet-1K, we randomly resize and crop all images to a resolution of 224×224, and apply random horizontal flip, normalization during training. We train ResNet with a batch size of 256, and adopt the cosine learning rate schedule with initial learning rate 0.1. The momentum and weight decay of SGD is set as 0.9 and 1$e$-4. SAM and SSAM use the same settings as above. The test images of both architectures are resized to 256×256 and then centerly cropped to 224×224. The perturbation magnitude $\rho$ is set to 0.07.

## 5.2 Experimental results

**Results on CIFAR10/CIFAR100.** We first evaluate our SSAM on CIFAR-10 and CIFAR100. The models we used are ResNet-18 [25] and WideResNet-28-10 [56]. The perturbation magnitude $\rho$ for SAM and SSAM are the same for a fair comparison. As shown in Table 1, ResNet18 with SSAM of 50% sparsity outperforms SAM of full perturbations. From Table 2, we can observe that the advantages of SSAM-F and SSAM-D on WideResNet28 are more significant, which achieve better performance than SAM with up to 95% sparsity. Note that the parameter size of WideResNet28 is much larger than that of ResNet-18, and CIFAR is often easy to overfit. In addition, even with very large sparsity, both SSAM-F and SSAM-D can still obtain competitive performance against SAM.

Table 1: Comparison between SGD, SAM and SSAM on CIFAR using ResNet-18

| Model | Optimizer | Sparsity | CIFAR10 | CIFAR100 | FLOPs[4] |
|---|---|---|---|---|---|
| | SGD | / | 96.07% | 77.80% | 1× |
| | SAM | 0% | 96.83% | 81.03% | 2× |
| | | 50% | **96.81%** (-0.02) | **81.24%** (+0.21) | 1.65× |
| | | 80% | 96.64% (-0.19) | 80.47% (-0.56) | 1.44× |
| | SSAM-F (Ours) | 90% | 96.75% (-0.08) | 80.02% (-1.01) | 1.36× |
| | | 95% | 96.66% (-0.17) | 80.50% (-0.53) | 1.33× |
| | | 98% | 96.55% (-0.28) | 80.09% (-0.94) | 1.31× |
| ResNet18 | | 99% | 96.52% (-0.31) | 80.07% (-0.96) | 1.30× |
| | | 50% | **96.87%** (+0.04) | **80.59%** (-0.44) | 1.65× |
| | | 80% | 96.76% (-0.07) | 80.43% (-0.60) | 1.44× |
| | SSAM-D (Ours) | 90% | 96.67% (-0.16) | 80.39% (-0.64) | 1.36× |
| | | 95% | 96.56% (-0.27) | 79.79% (-1.24) | 1.33× |
| | | 98% | 96.61% (-0.22) | 79.79% (-1.24) | 1.31× |
| | | 99% | 96.59% (-0.24) | 79.61% (-1.42) | 1.30× |

**Results on ImageNet.** Table 3 reports the result of SSAM-F and SSAM-D on the large-scale ImageNet-1K [8] dataset. The model we used is ResNet50 [25]. We can observe that SSAM-F and SSAM-D can maintain the performance with 50% sparsity. However, as sparsity ratio increases, SSAM-F will receive relatively obvious performance drops while SSAM-D is more robust.

**Training curves.** We visualize the training curves of SGD, SAM and SSAM in Fig. 2. The training curves of SGD are more jittery, while SAM is more stable. In SSAM, about half of gradient updates are the same as SGD, but its training curves are similar to those of SAM, suggesting the effectiveness.

---

[4]The FLOPs is theoretically estimated by considering sparse multiplication compared to SGD in Tables 1- 3.

Table 2: Comparsion between SGD, SAM and SSAM on CIFAR using WideResNet-28-10

| Model | Optimizer | Sparsity | CIFAR10 | CIFAR100 | FLOPs |
|---|---|---|---|---|---|
| | SGD | / | 97.11% | 81.93% | 1× |
| | SAM | 0% | 97.48% | 84.20% | 2× |
| WideResNet28-10 | SSAM-F (Ours) | 50% | **97.71%** (+0.23) | **85.16%** (+0.96) | 1.65× |
| | | 80% | 97.67% (+0.19) | 84.57% (+0.37) | 1.44× |
| | | 90% | 97.47% (-0.01) | 84.76% (+0.56) | 1.36× |
| | | 95% | 97.42% (-0.06) | 84.17% (-0.03) | 1.33× |
| | | 98% | 97.32% (-0.16) | 83.85% (-0.35) | 1.31× |
| | | 99% | 97.59% (+0.11) | 84.00% (-0.20) | 1.30× |
| | SSAM-D (Ours) | 50% | 97.70% (+0.22) | **84.99%** (+0.79) | 1.65× |
| | | 80% | **97.72%** (+0.24) | 84.36% (+0.16) | 1.44× |
| | | 90% | 97.53% (+0.05) | 84.16% (-0.04) | 1.36× |
| | | 95% | 97.52% (+0.04) | 83.66% (-0.54) | 1.33× |
| | | 98% | 97.30% (-0.18) | 83.30% (-0.90) | 1.31× |
| | | 99% | 97.27% (-0.25) | 84.16% (-0.04) | 1.30× |

Table 3: Comparsion between SGD, SAM and SSAM on ImageNet-1K using ResNet50.

| Model | Optimizer | Sparsity | ImageNet | FLOPs |
|---|---|---|---|---|
| | SGD | / | 76.67% | 1× |
| | SAM | 0% | 77.25% | 2× |
| ResNet50 | SSAM-F (Ours) | 50% | **77.31%** (+0.06) | 1.65× |
| | | 80% | 76.81% (-0.44) | 1.44× |
| | | 90% | 76.74% (-0.51) | 1.36× |
| | SSAM-D (Ours) | 50% | **77.25%** (-0.00) | 1.65× |
| | | 80% | 77.00% (-0.25) | 1.44× |
| | | 90% | 77.00% (-0.25) | 1.36× |

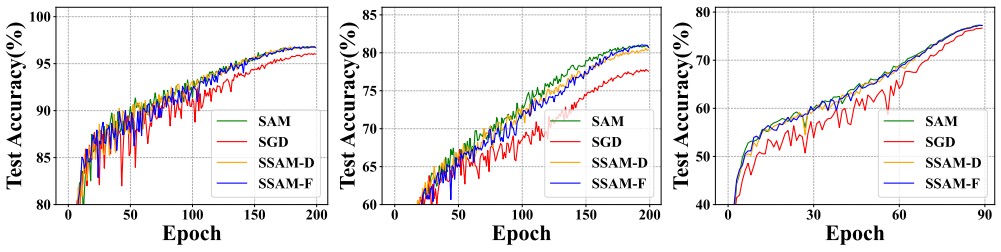

(a) ResNet18 on CIFAR10    (b) ResNet18 on CIFAR100    (c) ResNet50 on ImageNet-1K

Figure 2: The training curves of SGD, SAM and SSAM. The sparsity of SSAM is 50%.

**Sparsity *vs.* Accuracy.** We report the effect of sparsity ratios in SSAM, as depicted in Fig. 3. We can observe that on CIFAR datasets, the sparsities of SSAM-F and SSAM-D pose little impact on performance. In addition, they can obtain better accuracies than SAM with up to 99% sparsity. On the much larger dataset, *i.e.*, ImageNet, a higher sparsity will lead to more obvious performance drop.

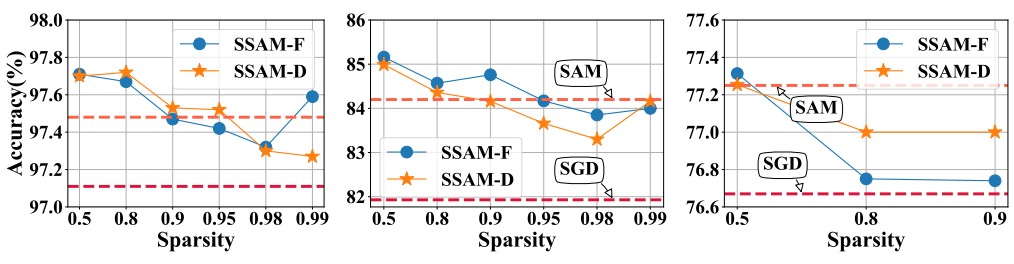

(a) WideResNet on CIFAR10.    (b) WideResNet on CIFAR100.    (c) ResNet50 on ImageNet.

Figure 3: Accuarcy *v.s.* sparsity on CIFAR10, CIFAR100 and ImageNet datasets.

### 5.3 SSAM with better generalization

**Visualization of landscape.** For a more intuitive comparison between different optimization schemes, we visualize the training loss landscapes of ResNet18 optimized by SGD, SAM and SSAM as shown in Fig. 4. Following [36], we sample $50 \times 50$ points in the range of $[-1, 1]$ from random "filter normalized" [36] directions, *i.e.*, the $x$ and $y$ axes. As shown in Fig. 4, the landscape of SSAM is flatter than both SGD and SAM, and most of its area is low loss (blue). This result indicates that SSAM can smooth the loss landscape notably with sparse perturbation, and it also suggests that the complete perturbation on all parameters will result in suboptimal minima.

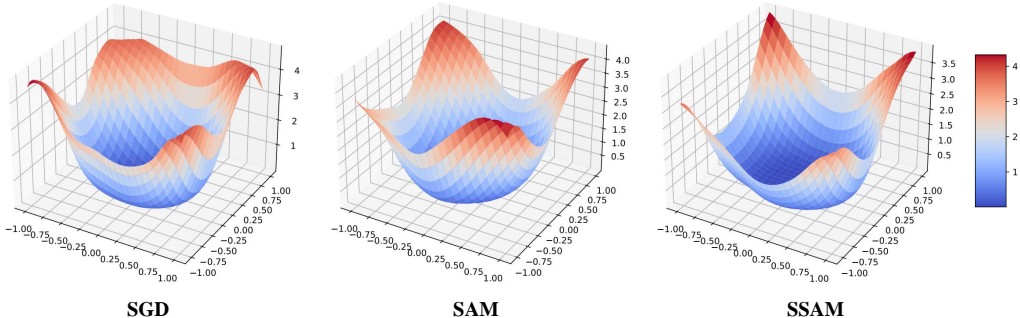

Figure 4: Training loss landscapes of ResNet18 on CIFAR10 trained with SGD, SAM, SSAM.

**Hessian spectra.** In Fig. 5, we report the Hessian spectrum to demonstrate that SSAM can converge to a flat minima. Here, we also report the ratio of dominant eigenvalue to fifth largest ones, *i.e.*, $\lambda_1/\lambda_5$, used as the criteria in [17, 29]. We approximate the Hessian spectrum using the Lanczos algorithm [21] and illustrate the Hessian spectra of ResNet18 using SGD, SAM and SSAM on CIFAR10. From this figure, we observe that the dominant eigenvalue $\lambda_1$ with SSAM is less than SGD and comparable to SAM. It confirms that SSAM with a sparse perturbation can still converge to the flat minima as SAM does, or even better.

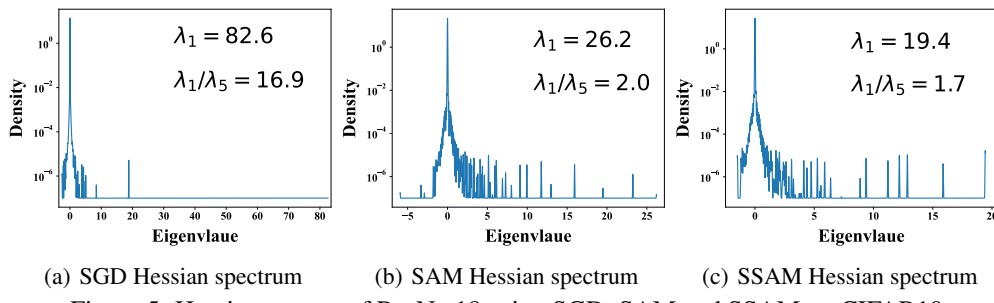

(a) SGD Hessian spectrum      (b) SAM Hessian spectrum      (c) SSAM Hessian spectrum

Figure 5: Hessian spectra of ResNet18 using SGD, SAM and SSAM on CIFAR10.

## 6 Conclusion

In this paper, we reveal that the SAM gradients for most parameters are not significantly different from the SGD ones. Based on this finding, we propose an efficient training scheme called Sparse SAM, which is achieved by computing a sparse perturbation. We provide two solutions for sparse perturbation, which are based on Fisher information and dynamic sparse training, respectively. In addition, we also theoretically prove that SSAM has the same convergence rate as SAM. We validate our SSAM on extensive datasets with various models. The experimental results show that retaining much better efficiency, SSAM can achieve competitive and even better performance than SAM.

## Acknowledgement

This work is supported by the Major Science and Technology Innovation 2030 "Brain Science and Brain-like Research" key project (No. 2021ZD0201405), the National Science Fund for Distinguished Young Scholars (No. 62025603), the National Natural Science Foundation of China (No. U21B2037, No. 62176222, No. 62176223, No. 62176226, No. 62072386, No. 62072387, No. 62072389, and No. 62002305), Guangdong Basic and Applied Basic Research Foundation (No. 2019B1515120049), and the Natural Science Foundation of Fujian Province of China (No. 2021J01002).

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
