# OpenReview forum: "Make Sharpness-Aware Minimization Stronger: A Sparsified Perturbation Approach"
_NeurIPS.cc/2022/Conference — NeurIPS 2022 Accept_

### Official Review · Reviewer_kk3C · 2022-06-29

**Rating:** 4
**Confidence:** 4
**Soundness:** 2 fair
**Presentation:** 3 good
**Contribution:** 2 fair

**Summary:**

This paper presents a sparse version of sharpness-aware minimization (SAM). The proposed SSAM does not apply perturbation to all the model parameters, where the criteria is based on the Fisherman Information or dynamically generating a mask through training. The authors claim improved generalization and training speed over SAM.

**Questions:**

Please see the weaknesses for questions.

**Limitations:**

The authors haven't covered the limitations. Potential limitations are limited results on other architectures besides ConvNets like Transformers.

**Strengths And Weaknesses:**

- Strengths:
    - The idea and observation that not all the model weights need to be perturbed in SAM is generally novel. The authors also provide empirical justification that the difference between $g_{SAM}$ and $g_{SGD}$ is not very significant for lots of weights.
    - I like the provided training curves as they show that SSAM behaves very alike to SAM in terms of the training dynamics.


- Weaknesses:
    - I don't quite understand how the FLOPs reduction are calculated in Table 1, 2, 3. From the algorithm, SSAM still requires twice full back-propagation. The only difference is multiplying the mask with the perturbations. Can the authors explain the FLOPs and it would be better to show the clock time of the training process.
    - I don't quite understand why option 2 is faster than option 1 when generating the mask (the first sentence in Section 4.3). From my understanding, they both rely on the gradient scale and require twice gradient computations.
    - Can the authors further demonstrate that the weight they select to perturb coincide with those appear to the right half in Figure 1? Just want to confirm that the masking strategy is align with the gradient information.
    - A random masking baseline should be included to show the effectiveness of SSAM.

---

> ### Author Response · Authors · 2022-08-02
> **Author response for the Reviewer kk3c: Part II**
>
> - **Q2: I don't quite understand why option 2 is faster than option 1 when generating the mask (the first sentence in Section 4.3). From my understanding, they both rely on the gradient scale and require twice gradient computations.**
>
>   A2: The option 1, *i.e.*, SSAM-F, is based on Fisher Information, which needs an additional computation. SSAM-F requires sampling $N_F$ data from dataset and calculate their respective gradients through an additional backpropagation. The option 2, *i.e.*, SSAM-D is based on dynamic sparse training and the gradients used to generate the mask comes directly from the gradient in the iteration, which means SSAM-D does not need additional gradient computations.
>
> - **Q3:  Can the authors further demonstrate that the weight they select to perturb coincide with those appear to the right half in Figure 1? Just want to confirm that the masking strategy is align with the gradient information.**
>
>   A3: Thank you for this suggestion. We count the statistics of ratio, *i.e.*, $ratio=\log |(g_{SAM}-g_{SGD})/g_{SGD}|$, in SSAM. Using SAM, the parameters with ratio greater than 0 account for 19% of the perturbated parameters, and the SSAM's ratio is 21.28%.
>
> - **Q4:  A random masking baseline should be included to show the effectiveness of SSAM.**
>
>   A4: Thank you for pointing the absent of this ablation. Following your suggestion, We experimented the SSAM with a random mask, *i.e.*, the **last** line with an accuracy of **77.08%**.  We report the results as shown in the following table.
>
>   |Model|Dataset| Optimizer | Sparsity |Detail |Acc |
>   |:--:|:--:|:--:|:--:|:--:|:--:|
>   |ResNet50|ImageNet|SGD|/| baseline | 76.67%|
>   |ResNet50|ImageNet|SAM|0| baseline | 77.25%|
>   |ResNet50|ImageNet|SSAM|50%|SSAM-D/SSAM-F|77.31%/77.25%|
>   |ResNet50|ImageNet|SSAM(Random)| 50% |Random Mask | 77.08% |
>
>
> - **Q5:**  Potential limitations are limited results on other architectures besides ConvNets like Transformers.
>
>   A5: Thanks for this suggestions. We experiment the Vit-Tiny training  on CIFAR10. Limited to the long training time and hardware requirements, we only train Vit-Tiny by SSAM-F. The results are shown below. It should be noted that ViT-Tiny has achieved 98% accuracy in CIFAR10, and the 0.16% improvement of SSAM is hard to reach.
>
>     | Model | Opimizer | Acc |
>     | :--: | :--: | :--:|
>     |Vit-Tiny| AdamW | 97.81%(-0.21%) |
>     |Vit-Tiny| SAM | 98.02%(+0.00%) |
>     |Vit-Tiny| SSAM-F 50% | 98.18%(+0.16%) |
>
> > ### Reference
> > - [1] [Towards Fully Sparse Training: Information Restoration with Spatial Similarity](https://aaai-2022.virtualchair.net/poster_aaai376)
> > - [2] [Accelerating Sparsity in the NVIDIA Ampere Architecture](https://developer.download.nvidia.com/video/gputechconf/gtc/2020/presentations/s22085-accelerating-sparsity-in-the-nvidia-ampere-architecture%E2%80%8B.pdf)

---

> ### Author Response · Authors · 2022-08-02
> **Author response for the Reviewer kk3c: Part I**
>
> ### Dear Reviewer kk3C:
>
> Thank you for the constructive comments which help us improve the paper.  We have prepared our responses to each of your point.
>
> - **Q1: I don't quite understand how the FLOPs reduction are calculated in Table 1, 2, 3. From the algorithm, SSAM still requires twice full back-propagation. The only difference is multiplying the mask with the perturbations. Can the authors explain the FLOPs and it would be better to show the clock time of the training process.**
>
>   A1: Sorry for our unclear presentation, we counted the time used in an iteration, and the ratio of forward and backward in one iteration is about 3:7. Based on this, the FLOPs reduction would be $0.7\times s$. We claim that the computing a sparse gradient is not much faster than computing the dense one. Since the weight-wise sparse training is limited to the hardware, we do not achieve actual weight-wise sparse training speedup. At present, Xu et.al. [1] has implemented the relevant sparse matrix calculation based on NVIDIA's 2:4 sparse operator [2], which means the sparse operator is gradually supported.
>
>   We report the wall-clock of our weight-wise Sparse SAM in the following table. Due to the absent of actual speedup of weight-wise sparse SAM and the extra computation of Fisher Information, the training time is longer.
>
>   | Model | Datasets | Optimizer | Sparsity | Test Acc | Time for One Epoch (Speed up) |
>   | :--: | :--: | :--: | :--: | :--: |:--:|
>   |ResNet18|CIFAR10|SAM| / | 96.83% | 13.60s(x1) |
>   |ResNet18|CIFAR10|Weight-Wise SSAM-F| 1%  | 96.74% | 14.21s(x0.957) |
>   |ResNet18|CIFAR10|Weight-Wise SSAM-F| 5%  | 96.93% | 15.08s(x0.901) |
>   |ResNet18|CIFAR10|Weight-Wise SSAM-F| 10% | 96.69% | 14.60s(x0.931) |
>   |ResNet18|CIFAR10|Weight-Wise SSAM-F| 25% | 96.82% | 14.76s(x0.921) |
>   |ResNet18|CIFAR10|Weight-Wise SSAM-F| 50% | 96.81% | 15.55s(x0.874) |
>   |ResNet18|CIFAR10|Weight-Wise SSAM-F| 80% | 96.64% | 14.30s(x0.951) |
>   |ResNet18|CIFAR10|Weight-Wise SSAM-F| 90% | 96.75% | 14.43s(x0.942) |
>   |ResNet18|CIFAR10|Weight-Wise SSAM-F| 95% | 96.66% | 15.15s(x0.897) |
>   |ResNet18|CIFAR10|Weight-Wise SSAM-F| 98% | 96.55% | 14.42s(x0.943) |
>   |ResNet18|CIFAR10|Weight-Wise SSAM-F| 99% | 96.52% | 14.26s(x0.953) |
>
>   To achieve really speedup for our Sparse SAM-Fisher, we also conduct the block-wise sparse fisher training. We actually compute the sparse gradients in the gradient ascent (to find the perturbation). We implement the block-wise sprase fisher in Pytorch. Specifically, we treat the block, *e.g.*, a kernel of an convolution layer, as unit and use `requires_grad` API in Pytorch to stop gradient computation. According to Pytorch's automatic differentiation mechanism, a parameter will no longer calculate gradient if none of its previous parameters require gradients. In this case, our block-wise SSAM Fisher achieve actually speedup.
>
>   The wall-clock of block-wise sparse training is shown in the following table. The table shows that our block-wise sparse SAM-Fisher is able to actually accelerate the training and maintain comparable performance.
>
>   | Model | Datasets | Optimizer | Sparsity | Test Acc | Training for One Epoch(Speed up) |
>   | :--: | :--: | :--: | :--: | :--: |:--: |
>   |ResNet18|CIFAR10|SAM| / | 96.83% | 13.60s(x1) |
>   |ResNet18|CIFAR10|Block-Wise SSAM-F| 1%  | 96.93% | 13.74s(x0.989) |
>   |ResNet18|CIFAR10|Block-Wise SSAM-F| 5%  | 96.76% | 13.90s(x0.978) |
>   |ResNet18|CIFAR10|Block-Wise SSAM-F| 10% | 96.86% | 13.83s(x0.983) |
>   |ResNet18|CIFAR10|Block-Wise SSAM-F| 25% | 96.84% | 14.05s(x0.967) |
>   |ResNet18|CIFAR10|Block-Wise SSAM-F| 50% | 96.80% | 13.69s(x0.993) |
>   |ResNet18|CIFAR10|Block-Wise SSAM-F| 80% | 96.64% | 13.61s(x0.999) |
>   |ResNet18|CIFAR10|Block-Wise SSAM-F| 90% | 96.55% | 13.56s(x1.002) |
>   |ResNet18|CIFAR10|Block-Wise SSAM-F| 95% | 96.65% | 13.20s(x1.030) |
>   |ResNet18|CIFAR10|Block-Wise SSAM-F| 98% | 96.66% | 12.79s(x1.063) |
>   |ResNet18|CIFAR10|Block-Wise SSAM-F| 99% | 96.53% | 12.62s(x1.077) |

---

> ### Author Response · Authors · 2022-08-09
> **Response to Reviewer kk3C**
>
> Dear reviewer kk3C,
>
> We thank for your time and suggestions. We have responded to your questions in detail accordingly. Would you mind checking it and confirming if you have further questions? Because there is only few hours left for us to answer your questions.
>
> Best,
>
> Authors

---

### Official Review · Reviewer_BQW5 · 2022-07-07

**Rating:** 7
**Confidence:** 4
**Soundness:** 3 good
**Presentation:** 3 good
**Contribution:** 3 good

**Summary:**

This paper rethinks the indiscriminate perturbations of SAM on all parameters and claims that not all parameters are needed to be perturbed. The conclusion is supported by two observations --- (i) little difference between SGD and SAM gradients in most dimensions and (ii) more flatter without SAM in some dimensions. Inspired by these observations, this paper proposes to improve the effectiveness and efficiency of SAM via a sparse perturbation scheme (SSAM). Two realizations of SSAM are provided, namely SSAM-Fisher and SSAM-Dynamic. The convergence of SSAM is also discussed. Extensive experiments, including visual and numerical results, demonstrate the effectiveness and efficiency of SSAM.

**Questions:**

Please refer to the weakness part

**Limitations:**

N.A.

**Strengths And Weaknesses:**

Strength:

- The paper is well-written and organized.
- Interesting observations on the flatness of SGD and SAM. It is surprising that the difference between SGD and SAM gradients is little and flat dimensions will be flatter without SAM
- It is quite natural to apply fisher information and dynamic strategy to encourage sparsity
- The effectiveness of SSAM is comparable to SAM; the computational cost is clearly reduced.

Weakness:
- I am very interested in why SAM makes the originally flat gradients worse?
- The ArgTopK operation selects $k$ parameters to be perturbed. How sensitive is the performance of SSAM with respect to $k$ in a certain task? How sensitive is the value of $k$ in different tasks (if training SOTA classifiers)?

---

> ### Author Response · Authors · 2022-08-02
> **Author response for Reviewer BQW5**
>
> ### Dear Reviewer BQW5:
>
> Thank you for your review and affirmation of our work. We'll answer your questions one by one below. We are also very honored to share our understanding with you.
>
> - **Q1: I am very interested in why SAM makes the originally flat gradients worse?**
>
>   A1: Interesting and insightful question. SAM is more effective compared to SGD, which has been confirmed in many different tasks and different models [1,2,3]. Our conjecture is that not every dimension needs to be smoothed, *i.e.*, not each parameter needs to be perturbated, since most of them are already flat. From the experimental results that performance becomes better when the flat parameters are not disturbed, SAM does make the originally flat gradients worse compared to the SSAM. We will study why SAM makes the flat gradient worse in the future.
>
> - **Q2: The ArgTopK operation selects $k$ parameters to be perturbed. How sensitive is the performance of SSAM with respect to $k$ in a certain task? How sensitive is the value of $k$ in different tasks (if training SOTA classifiers)?**
>
>   A2: The $k$ is the number describing how many parameters of the model in total will be perturbed. Since the number of parameters of each model is different, we use sparsity ratio $s$, *i.e.*, the proportion of parameters that are not perturbed to all parameters, to determine $k$ : $k=(1 - s)\cdot |w|$, where the $|w|$ is the number of model parameters. We report our SSAM with different sparsity $s$ optimizing ResNet, WideResNet on CIFAR10, CIFAR100 and ImageNet in our original paper, *e.g.*, Table 1, Table 2 and Table 3.
>
> > ### Reference
> > - [1] [Sharpness-Aware Minimization for Efficiently Improving Generalization](https://arxiv.org/abs/2010.01412)
> > - [2] [When Vision Transformers Outperform ResNets without Pre-training or Strong Data Augmentations](https://arxiv.org/abs/2106.01548)
> > - [3] [Sharpness-Aware Minimization Improves Language Model Generalization](https://arxiv.org/abs/2110.08529)

---

> ### Author Response · Authors · 2022-08-09
> **Response to Reviewer BQW5**
>
> Dear Reviewer BQW5
>
> We thank for your time and suggestions. We have responded to your questions in detail accordingly. Would you mind checking it and confirming if you have further questions? Because there is only few hours left for us to answer your questions.
>
> Best,
>
> Authors

---

### Official Review · Reviewer_KeX3 · 2022-07-09

**Rating:** 6
**Confidence:** 4
**Soundness:** 4 excellent
**Presentation:** 3 good
**Contribution:** 3 good

**Summary:**

The problem addressed in this study is that the indiscriminate perturbation manipulated by SAM on all parameters is suboptimal. The authors give two interesting observations that: there is little difference between the updating gradients of SGD and SAM in most dimensions; and there exist certain parameters that will still be flatter in some dimensions even if SAM is not used.
Based on the observations, they propose an efficient and effective training scheme to obtain sparse perturbations by crafting a binary mask. This paper employs the Fisher Information and dynamic sparse training to select top-K most significant and relative parameters to craft a sparse binary mask for perturbation.
The authors verify the effectiveness of their proposed training scheme on CIFAR10/CIFAR100 and ImageNet-1K datasets. The experimental results demonstrate that even with very large sparsity, both SSAM-F and SSAM-D can still obtain competitive performance against SAM. The authors also conduct visualization of landscape and Hessian spectra to show better generalization of SSAM.


**Questions:**

1. The results shown only from sparsity 50% and only for ResNet model. Could the authors have more experimental results on different sparsity and models with different architectures?

2. Why does it appear that employing fewer perturbation in Table 1 is unable to enhance the performance, if according to your purpose, that you can find certain parameters that can make loss flatter without perturbation?

3. As stated in Section 3.2, “merely about 5%parameter space is sharp while the rest is flat”. Do you believe you have correctly identified the 5% parameters that do not require perturbation application through your methods and that experimental results demonstrate this?

I am willing to increase my rating if my questions could be my addressed properly.


**Limitations:**

The authors haven’t addressed the limitations and potential negative societal impact of this work. It would be better to have discussion on SSAM’s effectiveness of more different datasets and model structures, as well as the potential future work of SSAM.

**Strengths And Weaknesses:**

Strengths:

1. The motivation is transparent and derived from the experimental observations. The authors accurately identify the indiscriminate perturbation in previous sharpness-related works. The motivational analysis and experiments are persuasive to me.
2. The written is straightforward and easy to understand. In addition, the landscape visualization is also informative from a motivational standpoint.

Weakness:
1. The experimental settings are not sufficient as compared with SAM paper. It would be better to have further experiments and a few ablation studies to confirm the model-agnostic characteristic of this technique.
2. There is a misunderstanding between the definitions of sparsity in line 176. The sparsity s should be the percentage of parameters to be perturbed, i.e., s=1 means all the parameters should be perturbated. However, s=1 in the table 1 means no parameters will be perturbated. Could the authors check for the definition?
3. It would be more convincing if the authors could have more analysis of the experimental outcomes. In the section titled "Questions," several inquiries regarding experiments are presented.

---

> ### Author Response · Authors · 2022-08-02
> **Author response for Reviewer KeX3: Part II**
>
> - **Q3: The results shown only from sparsity 50% and only for ResNet model. Could the authors have more experimental results on different sparsity and models with different architectures?**
>
>   A3: Following your suggestions, we report more experimental results on different sparsity first, *e.g.* 1%, 5%, 10%, 25% on CIFAR and ResNet. We maintain the setting and only change the sparsity of SSAM and the results are shown in the following table. To facilitate comparison, we also report the sparsity greater than 50%, which have been reported in original paper.
>
>   | Model | Optimizer |Sparsity|Acc(CIFAR10)| Acc(CIFAR100)|
>   |:--:|:--:|:--:|:--:|:--:|
>   |ResNet18|SGD          |/      |   96.07%    |   77.80%    |
>   |ResNet18|SAM          | 0%    |   96.83%    |   81.03%    |
>   |ResNet18|SSAM-F/SSAM-D| 1%    |96.74%/96.71%|81.01%/80.90%|
>   |ResNet18|SSAM-F/SSAM-D| 5%    |96.93%/96.71%|81.47%/80.98%|
>   |ResNet18|SSAM-F/SSAM-D| 10%   |96.69%/96.72%|81.07%/80.97%|
>   |ResNet18|SSAM-F/SSAM-D| 25%   |96.82%/96.83%|80.95%/80.84%|
>   |ResNet18|SSAM-F/SSAM-D| 50%   |96.81%/96.87%|81.24%/80.59%|
>   |ResNet18|SSAM-F/SSAM-D| 80%   |96.64%/96.76%|80.47%/80.43%|
>   |ResNet18|SSAM-F/SSAM-D| 90%   |96.75%/96.67%|80.02%/80.39%|
>   |ResNet18|SSAM-F/SSAM-D| 95%   |96.66%/96.56%|80.50%/79.79%|
>   |ResNet18|SSAM-F/SSAM-D| 98%   |96.55%/96.61%|80.09%/79.79%|
>   |ResNet18|SSAM-F/SSAM-D| 99%   |96.52%/96.59%|80.07%/79.61%|
>
>   For the different architectures, we conduct more experiment on CIFAR with VGG, and test the Vision-Transformer on CIFAR10. The detailed results could be found in Q1&A1.
>
> - **Q4: Why does it appear that employing fewer perturbation in Table 1 is unable to enhance the performance, if according to your purpose, that you can find certain parameters that can make loss flatter without perturbation?**
>
>   A4: We count the ratio, *i.e.*, $ratio=\log |(g_{SAM}-g_{SGD})/g_{SGD}|$,of the parameters selected by SSAM for perturbation. Using SAM, the parameters with ratio greater than 0 account for 19% of the perturbated parameters, and the SSAM has 21.28%.
>
> - **Q5: As stated in Section 3.2, “merely about 5% parameter space is sharp while the rest is flat”. Do you believe you have correctly identified the 5% parameters that do not require perturbation application through your methods and that experimental results demonstrate this?**
>
>   A5: We agree your opinion, we do not find the rigorous statistics like 5% ourselves. This statement comes from the reference https://arxiv.org/pdf/1609.04836.pdf, as a quote from the description in the fourth and fifth lines on page 7 of this paper, "we observe that it rises steeply only along a small dimensional subspace (e.g. 5% of the whole space); on most other directions, the function is relatively flat.". Motivated by the above discovery, we investigate the difference of gradient calculated by SGD and SAM with various models and datasets, *i.e.*, $log|(g_{sam}-g_{sgd})/g_{sgd}|$. As we can see in Fig.1 in our original paper, the gradient calculated by SGD for most parameters is not much different from that of SAM.
>
> Hope our answers resolve your concerns. If you have further confusion, we always welcome further discussion.

---

> > ### Comment · Reviewer_KeX3 · 2022-08-06
> > **Thanks for authors' response**
> >
> > Thanks for the response. The authors have clearly addressed my questions and concerns, I increase my score to 6.

---

> ### Author Response · Authors · 2022-08-02
> **Author response for Reviewer KeX3: Part I**
>
> ### Dear Reviewer KeX3:
>
> Thank you for your suggestions. Please see our follow-up responses. We hope the responses can further address your concerns. We always welcome further discussions.
>
> - **Q1: The experimental settings are not sufficient as compared with SAM paper. It would be better to have further experiments and a few ablation studies to confirm the model-agnostic characteristic of this technique.**
>
>   A1: Thanks for your suggestion. We conduct more experiments and ablations below.
>
>   - **1.** To confirm the effectness of our SSAM, we try to (1) randomly completely determine whether the weights will be perturbed or not, corresponding to the 4-th row with 77.08% acc, (2) random drop the weights for SSAM-D, corresponding to the 5-th row wi+th 77.08% acc, (3) drop the sharpnest weights, *i.e.*, weights with the smallest gradients, corresponding to the 6-th row with 76.68%. We also show the results reported in original paper in the first three lines. Recall the performance of SSAM-D, which drop the flattest weights is 77.25%. From the following table, we can see that drop the sharpnest weights are suboptimal for SAM and get a bad performance.
>
>     |Model|Dataset| Optimizer | Sparsity |Detail |Acc |
>     |:--:|:--:|:--:|:--:|:--:|:--:|
>     |ResNet50|ImageNet|SGD|/| baseline | 76.67%|
>     |ResNet50|ImageNet|SAM|0| baseline | 77.25%|
>     |ResNet50|ImageNet|SSAM|50%|SSAM-D/SSAM-F|77.31%/77.25%|
>     |ResNet50|ImageNet|SSAM(Random)| 50% |Random Mask | 77.08% |
>     |ResNet50|ImageNet|SSAM-D | 50% | Random Drop| 77.08% |
>     |ResNet50|ImageNet|SSAM-D | 50% | Drop Sharpnest | 76.68% |
>
>
>   - **2.** To confirm the model-agnostic characterisitic of the purposed method, we also test the VGG-style arch on CIFAR. Following the [https://github.com/weiaicunzai/pytorch-cifar100](https://github.com/weiaicunzai/pytorch-cifar100), we test our SSAM training the VGG11-BN on CIFAR, and the results are shown in the following table. For training as SAM and SSAM, we set $\rho=0.05$ for CIFAR10. The results are shown in the following table.
>
>     |Model|Dataset| Optimizer | Sparsity| Acc |
>     |:--:|:--:|:--:|:--:| :--:|
>     |VGG11-BN|CIFAR10|SGD   | /  | 93.42% |
>     |VGG11-BN|CIFAR10|SAM   | 0%  | 93.87% |
>     |VGG11-BN|CIFAR10|SSAM-F/SSAM-D| 1% | 93.85%/93.76% |
>     |VGG11-BN|CIFAR10|SSAM-F/SSAM-D| 5% | 94.01%/93.78% |
>     |VGG11-BN|CIFAR10|SSAM-F/SSAM-D| 10%| 93.92%/93.70% |
>     |VGG11-BN|CIFAR10|SSAM-F/SSAM-D| 25%| 93.88%/93.86% |
>     |VGG11-BN|CIFAR10|SSAM-F/SSAM-D| 50%| 94.03%/93.79% |
>     |VGG11-BN|CIFAR10|SSAM-F/SSAM-D| 80%| 93.83%/93.95% |
>     |VGG11-BN|CIFAR10|SSAM-F/SSAM-D| 90%| 93.76%/93.85% |
>     |VGG11-BN|CIFAR10|SSAM-F/SSAM-D| 95%| 93.77%/93.48% |
>     |VGG11-BN|CIFAR10|SSAM-F/SSAM-D| 98%| 93.54%/93.54% |
>     |VGG11-BN|CIFAR10|SSAM-F/SSAM-D| 99%| 93.47%/93.33% |
>
>   - **3.** To confirm the model-agnostic characterisitic of our SSAM, we plan to conduct the experiments training Vision-Transformer on CIFAR10. Limited to the long training time and hardware requirements, we only train Vit-Tiny by SSAM-F. The results are shown below. It should be noted that ViT-Tiny has achieved 98% accuracy on CIFAR10, and the 0.16% improvement of SSAM is very significant.
>
>     | Model | Opimizer | Acc |
>     | :--: | :--: | :--:|
>     |Vit-Tiny| AdamW | 97.81%(-0.21%) |
>     |Vit-Tiny| SAM | 98.02%(+0.00%) |
>     |Vit-Tiny| SSAM-F | 98.18%(+0.16%) |
>
>
> - **Q2: There is a misunderstanding between the definitions of sparsity in line 176. The sparsity s should be the percentage of parameters to be perturbed, i.e., s=1 means all the parameters should be perturbated. However, s=1 in the table 1 means no parameters will be perturbated. Could the authors check for the definition?**
>
>   A2: Thanks for pointing out this issue. We addmit the original presentation has a typo. The sparsity $s$ means *how many parameters do not calculate their perturbation?*, *i.e.*, $s=1$ means no parameters will be perturbated and $s=0$ means all parameters will be perturbated. We will fix our typo in the original paper in revsion.

---

### Official Review · Reviewer_ym8S · 2022-07-12

**Rating:** 5
**Confidence:** 4
**Soundness:** 2 fair
**Presentation:** 3 good
**Contribution:** 3 good

**Summary:**

The paper proposes to make the recent Sharpness-Aware Minimization (SAM), which tries to find flatter minima (which tend to generalize better) more efficient. Vanilla SAM incurs a 2x compute as 2 gradient computations (one for the perturbation and one at the perturbation) are needed at each training step. The paper proposes sparsifying SAM (SSAM) by only applying SAM on some fraction $s$ of the model weights. Which $s$ fraction is determined either in a cheap greedy fashion (SSAM-D) or via Fisher Info (SSAM-F). Non-convex analysis is done showing that SSAM converges at the same rate as SAM. SSAM is shown to perform nearly as well as SAM on benchmark tasks like CIFAR10, CIFAR100, and ImageNet using a ResNet model of modest size.


**Questions:**

* Ablations would greatly strengthen the paper.
* More experiments / datasets would also help.

**Limitations:**

Yes

**Strengths And Weaknesses:**

Strengths:
* The paper is well-motivated -- SAM is a powerful method but its 2x performance hit may hinder its widespread adoption and this paper shows how it can be sped up with little loss in performance.
* Paper is clear, easy-to-follow, and mostly free of typos.
* Theoretical analysis supports the proposed method.

Weaknesses:
* Although SSAM does in theory reduce SAM's FLOPS, it's unclear whether it would actually help in practice. Due to the nature of how gradients are computed in practice, computing gradients for a sparse subset of weights is often not much faster than computing the full set, especially when the subset is 50% and the weights could be from any of the layers. While the paper shows theoretical FLOPS, it does not have any numbers on actual wall-clock speed, etc, though I do understand that reporting wall-clock speed has some issues of its own.
* Key ablations missing. What happens if you invert the mask? For SSAM-D, instead of the dropping the flattest weights from the mask, drop the sharpest weights. Furthermore, what if a random subset is dropped?
* What's the influence of m-sharpness (doing m perturbations and averaging the gradients at the perturbations -- see original paper) on SSAM?
* Another strategy to reduce compute is using a 20% subset of the minibatch for the ascent gradient calculation (e.g. to find the perturbation). This strategy reduces the FLOPs significantly. How does SSAM compare to SAM when this approximation is used? Reference: https://arxiv.org/abs/2110.08529.
* The claim that SSAM can outperform SAM is weak: looking at Table 1, cifar100 95% and 98% have 0.40% gap for SSAM-F, which is larger than SSAM-F 50%'s improvement over SAM, suggesting the claimed improvement may just be noise.
* Tables start sparsity at 50%. Interested in seeing 1%, 5%, 10%,.. etc. At low sparsities it should be nearly identical to SAM.

---

> ### Author Response · Authors · 2022-08-02
> **Author response for Reviewer ym8S: Part III**
>
> - **Q6: Tables start sparsity at 50%. Interested in seeing 1%, 5%, 10%,.. etc. At low sparsities it should be nearly identical to SAM.**
>
>   A6: Great idea. Following your suggestions, we report more experimental results with different sparsity , *e.g.*, 1%, 5%, 10%, 25% on CIFAR and ResNet18 in the following table. For the convenience of comparison, we also show the existing results in the original paper.
>
>   | Model | Optimizer |Sparsity|Acc(CIFAR10)| Acc(CIFAR100)|
>   |:--:|:--:|:--:|:--:|:--:|
>   |ResNet18|SGD          |/      |   96.07%    |   77.80%    |
>   |ResNet18|SAM          | 0%    |   96.83%    |   81.03%    |
>   |ResNet18|SSAM-F/SSAM-D| 1%    |96.74%/96.71%|81.01%/80.90%|
>   |ResNet18|SSAM-F/SSAM-D| 5%    |96.93%/96.71%|81.47%/80.98%|
>   |ResNet18|SSAM-F/SSAM-D| 10%   |96.69%/96.72%|81.07%/80.97%|
>   |ResNet18|SSAM-F/SSAM-D| 25%   |96.82%/96.83%|80.95%/80.84%|
>   |ResNet18|SSAM-F/SSAM-D| 50%   |96.81%/96.87%|81.24%/80.59%|
>   |ResNet18|SSAM-F/SSAM-D| 80%   |96.64%/96.76%|80.47%/80.43%|
>   |ResNet18|SSAM-F/SSAM-D| 90%   |96.75%/96.67%|80.02%/80.39%|
>   |ResNet18|SSAM-F/SSAM-D| 95%   |96.66%/96.56%|80.50%/79.79%|
>   |ResNet18|SSAM-F/SSAM-D| 98%   |96.55%/96.61%|80.09%/79.79%|
>   |ResNet18|SSAM-F/SSAM-D| 99%   |96.52%/96.59%|80.07%/79.61%|
>
> > ### Reference
> > - [1] [Towards Fully Sparse Training: Information Restoration with Spatial Similarity](https://aaai-2022.virtualchair.net/poster_aaai376)
> > - [2] [Accelerating Sparsity in the NVIDIA Ampere Architecture](https://developer.download.nvidia.com/video/gputechconf/gtc/2020/presentations/s22085-accelerating-sparsity-in-the-nvidia-ampere-architecture%E2%80%8B.pdf)
> > - [3] [Efficient Sharpness-aware Minimization for Improved Training of Neural Networks](https://arxiv.org/abs/2110.03141)

---

> ### Author Response · Authors · 2022-08-02
> **Author response for Reviewer ym8S: Part II**
>
> - **Q2: Key ablations missing. What happens if you invert the mask? For SSAM-D, instead of the dropping the flattest weights from the mask, drop the sharpest weights. Furthermore, what if a random subset is dropped?**
>
>   A2: Following your suggestion, we performed more ablations, including (1) Sparse SAM, whose mask is generated completely random, shown in the 4-th row corresponding to "SSAM(Random)" as "Optimizer" and "/" as "Drop Criterion"; (2) SSAM-D, which drops a random subset, shown in the 5-th row corresponding to "SSAM-D" as "Optimizer" and "Random" as "Drop Criterion"; (3) SSAM-D, which drops the sharpest weights, shown in the 6-th row corresponding to "SSAM-D" for Optimizer and "Sharpnest weights" for Drop Criterion. In ablation, we use ImageNet as dataset and ResNet50 as model with 0.5 sparsity. The results of ablation are shown below. For the convenience of comparison, we also show the results in the original paper in the following Table, *i.e.*, the first three lines and the 7-th line.
>
>   | Optimizer | Drop Criterion |  Acc   |
>   |   :---:   |     :---:      |  :-:   |
>   |    SGD    |       /        | 76.67% |
>   |    SAM    |       /        | 77.25% |
>   |  SSAM-F   |       /        | 77.31% |
>   |  SSAM(Random)  |  /        | 77.08% |
>   | SSAM-D | Random            | 77.08% |
>   | SSAM-D | Sharpest weights  | 76.68% |
>   | SSAM-D | Flattest weights  | 77.25% |
>
>   From the above table, we can see that performance of SSAM-D drops a lot when SSAM-D drops the sharpest weights, even worse than SSAM-D dropping randomly, which is consistent with our conjecture.
>
> - **Q3: What's the influence of m-sharpness (doing m perturbations and averaging the gradients at the perturbations -- see original paper) on SSAM?**
>
>   A3: Thanks for this suggestion. Similar to SAM, we use a small ResNet on CIFAR10 using SSAM 50% with a range of values of $m$. The results are reported below.
>
>   |Optimizer|$m$| Acc |
>   |:--:|:--:|:--:|
>   |SSAM-D   |1  |94.89%|
>   |SSAM-D   |4  |94.96%|
>   |SSAM-D   |16 |95.21%|
>   |SSAM-F   |1  |94.60%|
>   |SSAM-F   |4  |94.95%|
>   |SSAM-F   |16 |95.35%|
>
> - **Q4: Another strategy to reduce compute is using a 20% subset of the minibatch for the ascent gradient calculation (e.g. to find the perturbation). This strategy reduces the FLOPs significantly. How does SSAM compare to SAM when this approximation is used? Reference: https://arxiv.org/abs/2110.08529.**
>
>   A4: Insightful question. We run the experiment which takes the 25% subset of minibatch to compute the perturbation with 50% sparsity. We keep the original settings unchanged except that only a part of minibatch is used to calculate the perturbation. We use ResNet18 on CIFAR10 in this part and the results are shown below.
>
>   | Optimizer | Data for Ascent Gradient |  Acc   | Training for One Epoch(Speedup)
>   |   :---:   |          :---:           |  :-:   | :--: |
>   |    SAM    |        100% minibatch    | 96.83% |  13.60s(x1)    |
>   |   SSAM-F  |   100% minibatch         | 96.81% |  15.55s(x0.87) |
>   |   SSAM-D  |   100% minibatch         | 96.87% |  14.67s(x0.92)
>   |    SAM    |   25% minibatch          | 96.55% |  12.56s(x1.08) |
>   |   SSAM-F  |   25% minibatch          | 96.57% |  13.38s(x1.01) |
>   |   SSAM-D  |   25% minibatch          | 96.66% |  13.47s(x1.009)|
>
>   Since the data to be computed is smaller, the training time decreased. However, we find a performance drop when taking 25% subset to compute the perturbation.
>
> - **Q5: The claim that SSAM can outperform SAM is weak: looking at Table 1, cifar100 95% and 98% have 0.40% gap for SSAM-F, which is larger than SSAM-F 50%'s improvement over SAM, suggesting the claimed improvement may just be noise.**
>
>   A5: Since we have finetuned the experimental setting of SAM on CIFAR100 carefully, *e.g.*, 81.03%(SAM in ours) *v.s.* 80.17%(SAM in [3]), the 0.21% improvement of SSAM-F 50% is hard to achieve. As for the high sparsity, *i.e.*, 95% and 98%, the model is closer to being optimized by SGD. The robustness of high sparsity is different from that of 50%. We rerun the ResNet18 with SSAM-F 50% on CIFAR100, and get 81.18%, 80.95% and 81.21% as accuracy.

---

> ### Author Response · Authors · 2022-08-02
> **Author response for Reviewer ym8S: Part I**
>
> ### Dear Review ym8S:
>
> Thank you for your all valuable suggestions and meticulous comments. We will incorporate your suggestions in the revision. Below we respond to your key concerns point by point. Please let me know if there are further more questions.
>
> - **Q1: Although SSAM does in theory reduce SAM's FLOPS, it's unclear whether it would actually help in practice. Due to the nature of how gradients are computed in practice, computing gradients for a sparse subset of weights is often not much faster than computing the full set, especially when the subset is 50% and the weights could be from any of the layers. While the paper shows theoretical FLOPS, it does not have any numbers on actual wall-clock speed, etc, though I do understand that reporting wall-clock speed has some issues of its own.**
>
>   A1: We agree that the computing a sparse gradient is not much faster than computing the dense one. Since the weight-wise sparse training is limited to the hardware, we do not achieve actual weight-wise sparse speedup. At present, Xu et.al. [1] has implemented the relevant sparse matrix calculation based on NVIDIA's 2:4 sparse operator [2], which means the sparse operator is gradually supported.
>
>   We report the wall-clock of our weight-wise Sparse SAM in the following table. Due to the absent of actual speedup of weight-wise sparse SAM and the extra computation of Fisher Information, the training time is longer.
>   | Model | Datasets | Optimizer | Sparsity | Test Acc | Time for One Epoch (Speed up) |
>   | :--: | :--: | :--: | :--: | :--: |:--:|
>   |ResNet18|CIFAR10|SAM| / | 96.83% | 13.60s(x1) |
>   |ResNet18|CIFAR10|Weight-Wise SSAM-F| 1%  | 96.74% | 14.21s(x0.957)  |
>   |ResNet18|CIFAR10|Weight-Wise SSAM-F| 5%  | 96.93% | 15.08s(x0.901) |
>   |ResNet18|CIFAR10|Weight-Wise SSAM-F| 10% | 96.69% | 14.60s(x0.931) |
>   |ResNet18|CIFAR10|Weight-Wise SSAM-F| 25% | 96.82% | 14.76s(x0.921) |
>   |ResNet18|CIFAR10|Weight-Wise SSAM-F| 50% | 96.81% | 15.55s(x0.874) |
>   |ResNet18|CIFAR10|Weight-Wise SSAM-F| 80% | 96.64% | 14.30s(x0.951) |
>   |ResNet18|CIFAR10|Weight-Wise SSAM-F| 90% | 96.75% | 14.43s(x0.942) |
>   |ResNet18|CIFAR10|Weight-Wise SSAM-F| 95% | 96.66% | 15.15s(x0.897) |
>   |ResNet18|CIFAR10|Weight-Wise SSAM-F| 98% | 96.55% | 14.42s(x0.943) |
>   |ResNet18|CIFAR10|Weight-Wise SSAM-F| 99% | 96.52% | 14.26s(x0.953) |
>
>   To achieve really speedup for our Sparse SAM-Fisher, we also conduct the block-wise sparse fisher training. We actually compute the sparse gradients in the gradient ascent (to find the perturbation). Specifically, we implement the block-wise sprase fisher in Pytorch. We treat the block, *e.g.*, a kernel of an convolution layer, as unit and use `requires_grad` API in Pytorch to stop gradient computation. According to Pytorch's automatic differentiation mechanism, a parameter will no longer calculate gradient if none of its previous parameters require gradients. In this case, our block-wise SSAM Fisher achieve actually speedup.
>
>   The wall-clock of block-wise sparse training is shown in the following table. The table shows that our block-wise sparse SAM-Fisher is able to actually accelerate the training and maintain comparable performance.
>
>   | Model | Datasets | Optimizer | Sparsity | Test Acc | Training for One Epoch(Speed up) |
>   | :--: | :--: | :--: | :--: | :--: |:--: |
>   |ResNet18|CIFAR10|SAM| / | 96.83% | 13.60s(x1) |
>   |ResNet18|CIFAR10|Block-Wise SSAM-F| 1%  | 96.93% | 13.74s(x0.989) |
>   |ResNet18|CIFAR10|Block-Wise SSAM-F| 5%  | 96.76% | 13.90s(x0.978) |
>   |ResNet18|CIFAR10|Block-Wise SSAM-F| 10% | 96.86% | 13.83s(x0.983) |
>   |ResNet18|CIFAR10|Block-Wise SSAM-F| 25% | 96.84% | 14.05s(x0.967) |
>   |ResNet18|CIFAR10|Block-Wise SSAM-F| 50% | 96.80% | 13.69s(x0.993) |
>   |ResNet18|CIFAR10|Block-Wise SSAM-F| 80% | 96.64% | 13.61s(x0.999) |
>   |ResNet18|CIFAR10|Block-Wise SSAM-F| 90% | 96.55% | 13.56s(x1.002) |
>   |ResNet18|CIFAR10|Block-Wise SSAM-F| 95% | 96.65% | 13.20s(x1.030) |
>   |ResNet18|CIFAR10|Block-Wise SSAM-F| 98% | 96.66% | 12.79s(x1.063) |
>   |ResNet18|CIFAR10|Block-Wise SSAM-F| 99% | 96.53% | 12.62s(x1.077) |

---

> ### Author Response · Authors · 2022-08-09
> **Response to Reviewer ym8S**
>
> Dear Reviewer ym8S
>
> We thank for your time and suggestions. We have responded to your questions in detail accordingly. Would you mind checking it and confirming if you have further questions? Because there is only few hours left for us to answer your questions.
>
> Best,
>
> Authors

---

### Author Response · Authors · 2022-08-09
**Author Response**

We thank all reviewers for your time and suggestions, and we expect to have a further discussion. We have responded to your questions in detail accordingly. If you have further questions or concerns, we still reply before the end of the author-reviewer discussion.  Thank you very much for your review time and efforts.

---

> ### Comment · Reviewer_ym8S · 2022-08-09
> **Increased my score from 4 to 5.**
>
> In light of the very thorough author response, wherein all my questions were answered, I've increased my score from 4 to 5. Given the low signal to noise ratio in the experimental results, the added complexity of this method, and the fact that the sparsity cannot be leveraged too well given current hardware / training setups (as the authors mention), I hesitate to score the paper as a 6. I, however, commend the authors for the comprehensive author response.

---

> > ### Author Response · Authors · 2022-08-10
> > **Author Response**
> >
> > We thank you for your reply. Your suggestions make our paper better. Using Pytorch API "requires_grad", we extend our SSAM-F into Block-wise SSAM-F, i.e., the unperturbed weights do not compute the gradient and save the training time. The Block-wise SSAM-F achieves the comparable performance on CIFAR and the training time decreases. Compared to the Block-wise SSAM-F, the extra time for computing Fisher Information is negligible.
> >
> >   | Model | Datasets | Optimizer | Sparsity | Test Acc | Training for One Epoch(Speed up) |
> >   | :--: | :--: | :--: | :--: | :--: |:--: |
> >   |ResNet18|CIFAR10|SAM| / | 96.83% | 13.60s(x1) |
> >   |ResNet18|CIFAR10|Block-Wise SSAM-F| 1%  | 96.93% | 13.74s(x0.989) |
> >   |ResNet18|CIFAR10|Block-Wise SSAM-F| 5%  | 96.76% | 13.90s(x0.978) |
> >   |ResNet18|CIFAR10|Block-Wise SSAM-F| 10% | 96.86% | 13.83s(x0.983) |
> >   |ResNet18|CIFAR10|Block-Wise SSAM-F| 25% | 96.84% | 14.05s(x0.967) |
> >   |ResNet18|CIFAR10|Block-Wise SSAM-F| 50% | 96.80% | 13.69s(x0.993) |
> >   |ResNet18|CIFAR10|Block-Wise SSAM-F| 80% | 96.64% | 13.61s(x0.999) |
> >   |ResNet18|CIFAR10|Block-Wise SSAM-F| 90% | 96.55% | 13.56s(x1.002) |
> >   |ResNet18|CIFAR10|Block-Wise SSAM-F| 95% | 96.65% | 13.20s(x1.030) |
> >   |ResNet18|CIFAR10|Block-Wise SSAM-F| 98% | 96.66% | 12.79s(x1.063) |
> >   |ResNet18|CIFAR10|Block-Wise SSAM-F| 99% | 96.53% | 12.62s(x1.077) |

---

### Meta-Review · Area_Chair_e3gy · 2022-08-27

**Recommendation:** Accept
**Confidence:** Certain

**Metareview:**

This paper proposes a new scheme to improve computational efficiency of the SAM optimizer. The original SAM requires to asses the loss value at a perturbed point. The perturbation lives in the full parameter space. This paper shows that computing the perturbation in every direction is not necessary. By only selecting a sparse subset of the parameters to undergo a perturbation, the authors are able to maintain the original generalization performance of SAM while significantly reducing the number of FLOPS. The paper also presents a convergence analysis for the proposed scheme.
There was an active discussion between authors and reviewers, and authors provided very very thorough response to the questions and issues raised by the reviewers. As a result of this, multiple reviewers increased the original score. At the end, 3 out of 4 reviews are on the positive side, and the remaining one is a borderline reject.
In concordance with the majority of the reviewers, I believe improving the computational cost of SAM is of huge practical interest, and this paper is a step toward in direction. I recommend accept. As the thorough reply from the authors contains interesting details, I strongly suggest that authors include these details to their submission (maybe to the supplementary material?)


**Award:**

No

---

### Decision · Program_Chairs · 2022-09-14

Accept